# Association of Financial Distress and Monthly Income with Smoking During the COVID-19 Pandemic Recession in Thailand: A Nationwide Cross-Sectional Study

**DOI:** 10.3390/ijerph22081287

**Published:** 2025-08-18

**Authors:** Chutarat Sathirapanya, Wit Wichaidit, Vijj Kasemsup, Vasin Pipattanachat, Rassamee Chotipanvithayakul

**Affiliations:** 1Tobacco Control Research and Knowledge Management Center, Southern Node, Hat Yai 90110, Songkhla, Thailand; rassamee.s@psu.ac.th; 2Department of Family and Preventive Medicine, Faculty of Medicine, Prince of Songkla University, Hat Yai 90110, Songkhla, Thailand; 3Department of Epidemiology, Faculty of Medicine, Prince of Songkla University, Hat Yai 90110, Songkhla, Thailand; wit.w@psu.ac.th; 4Faculty of Medicine, Ramathibodi Hospital, Mahidol University, Bangkok 10400, Thailand; vijj.kas@mahidol.ac.th; 5Tobacco Control Research and Knowledge Management Center, Bangkok 10400, Thailand; hidedz99@gmail.com

**Keywords:** psychological stress, cigarette smoking, income, economic recession, COVID-19 pandemic

## Abstract

(1) Background: Psychological distress causes increased smoking frequency. Thus, financial distress (FD), a type of psychological distress, during the COVID-19 pandemic recession is possibly associated with increased smoking frequency. We studied the association between disrupted employment or earnings-associated FD and the number of cigarettes smoked daily, as well as the effects of pre-pandemic regular monthly incomes on the association. (2) Methods: We retrieved the collected data from a former nationwide and community-based study regarding the psycho-socio-economic characteristics of Thai people during the third wave of the COVID-19 pandemic in Thailand in 2021. The participants for this study were current smokers aged ≥ 18 years. General demographics, perceived FD, and pre-pandemic regular monthly incomes were analyzed. Descriptive statistics and multivariate logistic regression with sampling weight adjustments were the analyses used (*p* < 0.05). (3) Results: 849 current smokers were suitable for statistical analyses. We found that 664 (12.9%) of participants experienced FD, but it did not significantly affect smoking frequency. However, those who had FD and regularly earning ≥ THB 10,000 ($271.11 US) per month was significantly associated with increased daily cigarette use, after adjusting for age and sex (adjusted OR = 2.74; 95% CI = 1.18, 6.37, *p* = 0.020). (4) Conclusion: FD alone did not affect daily smoking frequency. Understanding the psycho-socio-economic factors is necessary for smoking control during the COVID-19 economic recession.

## 1. Introduction

Cigarette or tobacco smoking remains a pivotal public health problem worldwide. Many tobacco-related health hazards are well recognized. The WHO has endorsed an international agreement under the Framework Convention on Tobacco Control (FCTC) aimed at a 30% reduction in smoking prevalence globally [1]. However, psychological distress caused by personal life stress or socio-economic hardship provokes increased cigarette or tobacco use [2,3,4,5,6]. The recent COVID-19 pandemic resulted in a global health disaster, as well as a widespread economic breakdown. The term “COVID-19 pandemic recession” [7] was used to represent economic collapse during this global health crisis. Because nearly all enterprises, industries and social events were shut down, a high unemployment rate resulted in financial distress (FD), a type of psychological distress, which possibly resulted in increased smoking frequency for coping with the distress [8,9,10,11]. Increased cigarette or tobacco product use was found among unemployed people with severe psychological distress during the economic recession [8,12,13,14,15]. However, a study reported that a reduction in cigarette use, or “cutting back”, was applied to limit expenditures among smokers during the economic recession [14]. Also, higher daily cigarette use was found only among tobacco-dependent unemployed workers [9]. Therefore, the available studies concerning the association between smoking frequency and FD occurring during the COVID-19 pandemic recession remain controversial.

Smoking in Thailand has been restricted following the Tobacco Control Act 2017 to prevent both active and passive smokers from acquiring smoking-attributed diseases. The law restricts local production, importation, advertising, and sale to people aged under 18 years. Smoking in public venues, private or governmental offices, buses, or marketplaces, etc., was strictly prohibited in compliance with the act. The anti-smoking measures applied could decrease smoking prevalence from 43.7% to 34.7% in males and from 2.6% to 1.3% in females between 2004 and 2021 [16].

The third wave of the COVID-19 pandemic in Thailand lasted from January to December 2021. During that time, many Thai people faced financial difficulty, leading to FD. The study on smoking frequency among lower-income Thai smokers, a major proportion of current smokers in Thailand, who had FD during the COVID-19 pandemic recession was limited. As a pack of local cigarettes costs 65 to 105 THBs ($1.76 to 2.85 US), or approximately 1.5–2.5 h of the minimum wage [17], we proposed that the lower-income smokers would reduce their smoking frequency to limit their expenditure. Additionally, how the higher-income current smokers practiced, despite having FD, was studied as well. Understanding smoking behaviors during the pandemic recession will provide the essential data for health and economic mitigation.

## 2. Materials and Methods

### 2.1. Study Design and Setting

The data for statistical analysis in this study were from a nationwide survey of the psycho-socio-economic characteristics during the third wave of the COVID-19 pandemic in Thailand, which was conducted from June to September 2021.

### 2.2. Study Participants and Sampling Methods

The data collected in the nationwide survey 2021 were from people aged 15 years or older. A stratified sampling technique was applied for enrolment to include samples that would represent the entire nation. Initially, the randomly selected provinces for data collection were from the twelve health administration zones, following the administrative areas defined by the Ministry of Public Health of Thailand. Then, districts or subdistricts of the initially selected provinces were classified into urban or rural areas, and further randomly selected at a ratio of urban to rural areas of 1:3.

In this study, we calculated sampling weight to determine the required number of data for statistical analysis by dividing the number of participants aged 18 years and older (cigarettes were prohibited for purchasing among people age less than 18 years according to tobacco control law of Thailand) from each zone by the mid-year population of 18 years or older in 2020.

### 2.3. Study Instrument

The study instrument used in the nationwide survey 2021 was a structured questionnaire comprising multiple-choice and open-ended questions. It was developed by a consortium of investigators who were experts in the fields of public health, humanities, social sciences, and medicine from various universities in Thailand. The final questionnaire included 6 parts which had a total of 25 items: (1) General characteristics, (2) Number of cigarettes smoked during the pre-pandemic period, (3) Number of cigarettes smoked during the third wave of the COVID-19 pandemic, (4) Expenses in purchasing cigarettes and other tobacco products, (5) Knowledge regarding the adverse health effects of smoking during the COVID-19 pandemic, (6) Intention to quit or reduce frequency of cigarette smoking. The questionnaire was programmed onto a Google Form platform requiring 10 min for completion by self-administration. The developed questionnaire underwent the peer review process to confirm its integrity without any systematic assessments of the content validity or reliability applied (see Appendix A).

### 2.4. Experience of Financial Distress (FD)

Experience of FD due to disrupted employment or earnings during the COVID-19 pandemic recession in this study was defined when the participants answered to question No 5., “Employment status during the third wave of the COVID-19 pandemic”, with one of the following answers: “Reduced number of work days”, “Temporary suspension of employment”, “Laid off or the termination of business”, “Dismissed from work”, “Contract Expiration”, or “Business Closure”. If the participant answered, “Came to work as usual” or “Quit working”, they were considered to have no FD. We deemed that the last response represented an individual’s voluntary departure from work due to personal reasons.

### 2.5. Pre-Pandemic Regular Personal Monthly Income

The regular personal monthly income before the pandemic was asked through question No. 3 as “Q3. Regular monthly income .............. THB (please specify)”. We specified 10,000 THB ($271.11 US) as the cut-off point of high or low monthly income based on the average earnings used in the other nationwide study of monthly income among Thai workers [13].

### 2.6. Changes in Number of Daily Cigarettes Smoked

The participants reported the estimated number of cigarettes smoked before and during the COVID-19 pandemic recession. The results were used to classify the smokers into those who smoked more, stable, or fewer daily cigarettes during the time of the pandemic recession compared with the pre-pandemic period.

### 2.7. Data Collection and Management

The initial data received were cleaned and planned for data pooling using Microsoft Excel by the analyst (W.W.). An Excel file was then opened with the R statistical environment for data analysis.

### 2.8. Data Analyses

Descriptive statistics were used to display general demographics and study variables. We performed weighted analyses to provide estimates with a margin of error (standard errors) using the survey package in the R program [18]. The data were analyzed by descriptive univariate and bivariate analyses using weighted percentages (or weighted mean) and standard errors. The analysis was adjusted for the potential confounders of age and sex in the multivariate regression model. We also performed the Breslow–Day test of homogeneity using unweighted data.

### 2.9. Ethical Considerations

The study protocol was approved by the Human Research Ethics Committee, Faculty of Medicine, Prince of Songkla University (Code No. REC 67-100-9-1, date of approval 16 February 2024). We strictly followed the regulations stated in the 1964 Declaration of Helsinki and the relevant ethical practice guidelines in performing this research. The participants’ identifiable information was completely anonymous.

## 3. Results

A total of 4597 participants, mostly male, were collected in the database of the nationwide survey 2021. Among them, 42.4% were current smokers, and 56% earned less than 10,000 THB ($271.11 US) per month before the pandemic.

Among the 1826 current smokers, 849 were suitable for statistical analysis in this study; 65.6% of the participants maintained their smoking at the same level as the pre-pandemic period (Table 1). There was no difference in smoking frequency between those who had and did not have FD (Table 2).

When regular monthly income before the pandemic was considered, the study participants who had FD and earned 10,000 THB or more showed a significant association with an increase in the daily number of cigarettes smoked after adjusting for age and sex (Adjusted OR =2.74; 95% CI = 1.18, 6.37, *p* = 0.020) (Table 3).

## 4. Discussion

This study described the association between FD during the COVID-19 pandemic recession, FD and regular monthly income (before the pandemic), and changes in smoking frequency. We found that the current smokers who experienced FD did not significantly change their smoking frequencies during the COVID-19 pandemic recession compared with the pre-pandemic period. However, only those who earned ≥10,000 THB ($271.11 US) per month had significant increase in cigarettes smoked. The findings did not agree with our initial hypothesis that most smokers would reduce smoking to cut down on expenditures. FD during the pandemic recession did not affect the smoking frequency of the current smokers in this study. Although a study among the tobacco-dependent people reported higher smoking frequency during the pandemic [9], the current smokers in this study decided to maintain the same smoking frequency in response to their tobacco dependence while keeping household expenditure in balance as much as possible. Those who had FD but earned a higher income would smoke more than those who earned less because they could still afford to buy cigarettes. Hence, they were able to use smoking as a stress-coping behavior during the pandemic recession. A study from England showed a slow decline in the rate of smoking prevalence during the COVID-19 pandemic recession because of the contrary rise in smoking prevalence among younger people, while smoking prevalence among older people was reduced. Like the finding of the present study, the smokers at more advantaged socio-economic levels showed a pronouncedly slow decline in smoking rate [11]. Moreover, an immediate increase in daily smoking frequency was reported among the current smokers during the COVID-19 pandemic as well [15].

Various studies have confirmed that precariously employed or unemployed people experienced psychological consequences [6,19,20,21,22], especially depression, which was frequently found among men and head-of-household women [9]. Jahoda M, in 2019, proposed the latent deprivation model among unemployed people, highlighting that they would lose both ‘manifest functional employment (earning income)’ and ‘latent functional employment (time structure of a day, social contact, social activities, and perceived value by others)’ during job loss [23]. Both deprivations independently predicted negative mental health status [21]. Psychological distress, socio-economic disadvantage, and food insecurity were interactive risk factors of high smoking prevalence in the 2015 US Study of Income Dynamics. The study showed that although food insecurity alone was associated with higher smoking prevalence, only psychological distress became a significant predictor of smoking when food insecurity and inadequate household earnings were adjusted [5]. Therefore, psychological distress caused by whichever stressors can be a predictor of higher smoking frequency.

Although studies reported that unstable employment increased smoking frequency [24,25], the results reporting the association between FD and smoking frequency were controversial. A study in Thailand conducted during the same period found no significant association between FD and smoking frequency [13]. Moreover, no impact of regular income and psychological distress on the association between job loss and smoking cessation was found [26].

The Granular Intersection Thinking theory (GITT), a quantum mechanics theory, states that the functions of the macrosystem rely on the interactions of the microstructures composing the macrosystem. This concept could be applied to social, or economic system [27,28]. The ecosystem of human life is similar in that several composing factors interact with each other, i.e., health, socio-economic, knowledge, and psychological, as well as an individual’s cognitive statuses. When the composing factors are stable, the ecosystem operates normally. However, when either of them is disrupted or synchrony is lost, like during the COVID-19 pandemic recession, the ecosystem fails. Since humans possess cognitive and psychological capabilities to cope with encountered stressors, a selection of coping mechanisms will bring about a behavior, in which smoking will be practiced under stressful conditions.

In conclusion, there is possibly a complex interaction among work loss, psychological distress, coping mechanisms, and income on smoking frequency to cope with encountered economic crisis. To maintain the efficacy of smoking control measures for public health during an economic recession, various non-health-related factors should be collaboratively considered.

## 5. Strengths and Limitations

A strength of our study is that it is based on a nationwide study that systematically and randomly selected representatives covering every part of Thailand. However, due to its small sample size, data concerning alternative incomes (i.e., government-supported money, or self-reserved money) and duration and severity of the perceived FD were lacking; thus, the generalizability of the results could be weakened. Otherwise, reporting bias caused by social desirability or norms could lead to the underreporting of FD, personal monthly earnings, and the actual number of daily cigarettes, or the alternatives, smoked. Future research should evaluate the interaction among psychological reactions to FD, direct and indirect incomes, attitudes towards health, and the coping mechanisms selected to relieve FD, on smoking.

## 6. Conclusions

The current smokers in Thailand who were high earners and had FD reported a significantly increased number of cigarettes smoked during the third wave of the COVID-19 pandemic recession, while FD alone did not affect smoking frequency. The change in smoking frequency during the COVID-19 pandemic recession is possibly a complex interrelation among health, psychological, social, and economic factors.

## Figures and Tables

**Table 1 ijerph-22-01287-t001:** Characteristics of the study participants from the nationwide survey. Data were collected from June–September 2021.

Characteristic	Crude Count	Weighted Percent ± Standard Error, Unless Otherwise Noted
Sex (gender assigned at birth)	*n* = 4597	
Male	3114	69.9% ± 0.7%
Female	1483	30.1% ± 0.7%
Age in years, mean ± SE		41.4 ± 0.2
Age Groups	*n* = 4597	
Age 18 to 29 years	1324	28.5% ± 0.6%
Age 30 to 39 years	916	17.5% ± 0.6%
Age 40 to 49 years	890	19.6% ± 0.6%
Age 50 to 59 years	880	20.8% ± 0.6%
Age 60 years or older	587	13.7% ± 0.5%
Personal Monthly Income (THB) ($ US at $1 US = 36.89 THB)	*n* = 4590	(*n* = 4590)
Less than 5000 THB	1402	30.6% ± 0.7%
Between 5000 THB and 9999 THB	1193	25.4% ± 0.7%
Between 10,000 THB to 14,999 THB	597	12.1% ± 0.5%
Between 15,000 THB to 19,999 THB	452	9.5% ± 0.5%
Between 20,000 THB to 24,999 THB	289	6.4% ± 0.4%
Between 25,000 THB to 29,999 THB	199	4.5% ± 0.3%
30,000 THB or more	458	11.4% ± 0.5%
Financial Distress During the third wave COVID-19 Pandemic	*n* = 4597	
Did not experience financial distress	3933	87.1% ± 0.5%
Experienced financial distress	664	12.9% ± 0.5%
Smoking Status at the time of the Survey	*n* = 4553	
Never or former smokers	2727	57.6% ± 0.7%
Current smokers	1826	42.4% ± 0.7%
Smoking During the pandemic vs. pre-pandemic (among the current smokers only)	*n* = 849	
Smoked less during the pandemic compared to pre-pandemic	201	23.5% ± 1.5%
Smoked the same during the pandemic compared to pre-pandemic	563	65.6% ± 1.7%
Smoked more during the pandemic compared to pre-pandemic	85	10.9% ± 1.1%

**Table 2 ijerph-22-01287-t002:** Association between financial distress and changes in cigarette use during the third wave of the COVID-19 pandemic recession among study participants (weighted percent ± standard error) (*n* = 849 Thai adults).

Financial Distress	Smoked with Less Frequency	Smoked at the Same Frequency	Smoked with More Frequency	Crude OR (95% CI) for Smoking Less vs. the Same Frequency	Crude OR (95% CI) for Smoking More vs. the Same Frequency	Adj. OR * (95% CI) for Smoking Less vs. the Same Frequency	Adj. OR * (95% CI) for Smoking more vs. the Same Frequency
No (*n* = 717)	22.5% ± 1.6%	66.9% ± 1.8%	10.5% ± 1.2%	Reference	Reference	Reference	Reference
Yes (*n* = 132)	29.0% ± 4.1%	57.8% ± 4.5%	13.2% ± 3.2%	1.49 (0.96–2.32)	1.45 (0.78–2.68)	1.47 (0.94–2.29)	1.27 (0.68–2.38)
	*p* value			0.078	0.241	0.091	0.450

* Adjusted for sex and age of the study participants.

**Table 3 ijerph-22-01287-t003:** Association between financial distress and changes in cigarette use during the third wave of the COVID-19 pandemic recession among the study participants, stratified by income level (weighted percent ± standard error) (*n* = 843 Thai adults).

Financial Distress	Smoked with Less Frequency	Smoked at the Same Frequency	Smoked with More Frequency	Crude OR (95% CI) for Smoking Less vs. the Same Frequency	Crude OR (95% CI) for Smoking More vs. the Same Frequency	Adj. OR * (95% CI) for Smoking Less vs. the Same Frequency	Adj. OR * (95% CI) for Smoking More vs. the Same Frequency
Among those with a personal monthly income below THB 10,000 ($271.11 US) per month (*n* = 407)							
No (*n* = 306)	22.1% ± 2.5%	70.2% ± 2.7%	7.6% ± 1.6%	Reference	Reference	Reference	Reference
Yes (*n* = 101)	31.8% ± 4.8%	61.4% ± 5.1%	6.8% ± 2.8%	1.64 (0.97–2.78)	1.02 (0.38–2.71)	1.64 (0.97–2.79)	0.90 (0.32–2.52)
	*p* value			0.064	0.969	0.066	0.837
Among those with a personal monthly income of THB 10,000 ($271.11 US) or more per month (*n* = 436)							
No (*n* = 405)	23.0% ± 2.2%	64.4% ± 2.5%	12.7% ± 1.7%	Reference	Reference	Reference	Reference
Yes (*n* = 31)	21.1% ± 7.6%	47.9% ± 9.2%	31.0% ± 8.5%	1.24 (0.46–3.31)	3.29 (1.36–7.96)	1.12 (0.42–2.97)	2.74 (1.18–6.37)
	*p* value			0.671	0.009	0.823	0.020

* Adjusted for sex and age. Six participants who did not provide data regarding personal monthly income were excluded. Breslow–Day Test of Homogeneity for smoking less (Ref. same) *p*-value = 0.352 Breslow–Day Test of Homogeneity for smoking more (Ref. same) *p*-value = 0.034.

## Data Availability

The study data and methods were reported in the article. No study data were deposited in other data depository sources or websites.

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
