# Peer review of "Association of Financial Distress and Monthly Income with Smoking During the COVID-19 Pandemic Recession in Thailand: A Nationwide Cross-Sectional Study"

_ijerph, 2025, doi:10.3390/ijerph22081287_

Round 1

Reviewer 1 Report

Comments and Suggestions for Authors

This is a very well-constructed paper. My minor comments are below:

Lines 18-20: Claims are made in the first two sentences of background of the abstract, particularly increased cigarette use with psych distress and FD during COVID increasing smoking history. Please include references to peer-reviewed literature that evidences these statements.

Lines 40-71 - What is prevalence of smoking in the Thai population? What is the ecosystem for Thai smokers - i.e. smoke free places? Restrictions on products used? How expensive is an average pack of cigarettes in Thailand? More background on the daily life of cigarette use in Thailand would be helpful context for the findings of this study.

Line 41 - Please expand and explain more on tobacco related health hazards, especially in the context of this paper. Is the interest in psychological side-effects, withdrawal, cancer risk? Please include citations as well.

Line 48 - A minor point here, but the "Great Recession" classically refers to the 2008 US market crash initiated by the sub-prime mortgage lending, not the COVID-19 pandemic. The paper cited [8] does reference the Great Recession as occurring in 2008 as a comparator to the COVID induced downturn - recommend removing the reference to the Great Recession there.

Line 162 - Table 1. Please include an n column containing crude counts, as a comparator to the weighted percent.

Line 176-87. The proportional spending on cigarettes in the place of other needs in individuals who are experiencing FD is established in the literature, so the core finding here is not out of place and corroborates those other papers. The finding of this paper defends that heavy smokers would preferentially choose to cut other needs (i.e. leisure spending, sometimes even food) to keep smoking at the same frequency, which is a fascinating result and in line with results presented in many other papers. I would recommend including a citation or two that discusses that behavior, particularly on or near line 185.  

Lines 246-257. I was curious as to why so few respondents reported FD during the 3rd wave (12.9%). Is there something in the design that would lead to this small percentage, or are respondents less likely to self-report a stigmatized topic? Worth discussing in the limitations section.

Author Response

Dear reviewer,

Thank you very much for your suggestions and comments.

Please find the point-by-point responses to your comment below.

Regards,

Reviewer 2 Report

Comments and Suggestions for Authors

Dear Author,

Overall, the paper demonstrates potential and may eventually contribute meaningfully to the academic literature. However, there are several major issues that need to be addressed before it can reach that stage.

First, the current title is both redundant and unclear. I recommend revising it to make it more concise while accurately reflecting the study’s core focus.

Second, it is unclear what is meant by “modified by” in the manuscript. If the intended meaning is “moderated by,” the authors should clarify this throughout the text. Moreover, if moderation is implied, the study should include an interaction term between financial distress and income level in the regression model to test for non-additive effects. This is essential for accurately examining whether income level moderates the relationship between financial distress and cigarette smoking behavior.

Third, the presentation of results is currently confusing. Although logistic regression was employed, the tables do not clearly indicate the statistical significance levels of the findings, which is a standard requirement for reporting statistical analyses. Clearer and more transparent reporting of results is necessary to enhance the paper’s credibility and readability.

Fourth, the manuscript lacks sufficient theoretical grounding to justify the hypotheses, particularly regarding the proposed non-linear relationship between financial distress and cigarette use frequency as conditioned by income. The authors are encouraged to elaborate on the theoretical rationale underpinning this relationship. In particular, the Granular Interaction Thinking Theory (GITT) may offer a valuable lens to understand non-linear dynamics between psychological states and behavioral outcomes. For reference, please consult:

  • Vuong QH, Nguyen MH. (2024). Better economics for the Earth: A lesson from quantum and information theories. https://books.google.com/books?id=I50TEQAAQBAJ

Fifth, the current discussion and implications are relatively superficial. With the support from GITT, the authors can discuss that the increased smoking among higher-income individuals under FD may represent an emergent behavior arising from the intersection of stress exposure and the retained economic ability to afford stress-coping mechanisms. Moreover, the authors may expand the discussion to reason how the decision to smoke more (despite economic strain) reflects cognitive trade-offs: the short-term emotional relief outweighs long-term economic or health costs as long as the individual has the financial capability to make the tradeoff. GITT helps explain this context-dependent rationality.

Finally, the language needs to be thoroughly revised to avoid incorporating subjective expressions, such as “we believe.”

I hope you find these comments helpful in strengthening your manuscript. I look forward to reading a thoroughly revised version in due course.

Best regards,

Author Response

(The authors gave the same response as above.)

Reviewer 3 Report

Comments and Suggestions for Authors

Title: A cross-sectional nationwide study of the association between financial distress and daily cigarettes smoked, modified by regular monthly income during the economic recession in the third wave of the COVID-19 pandemic in Thailand

Overall:

This study has several ambiguities that make it difficult to evaluate, and the findings do not significantly advance the field. Sentence structure, grammar, etc. should be revisited, and proofreading is needed.

Abstract:
Rephrase to be more scientific language: “Financial distress (FD) during COVID-19 pandemic recession probably increased smoking frequency” – e.g., “Thus, financial distress (FD), one type of psychological distress, may be associated with increased smoking frequency.” I would guess that there is some evidence that FD is associated with smoking? If so, state that directly.

Methods – it wasn’t clear how FD was assessed and what the outcome was.

Results – rephrase for grammar/sentence structure: “However, those who had FD and regularly earned ≥THB 10,000 ($271.11 US) per month 30 was significantly associated with increased daily cigarettes use, after adjusting for age and 31 sex (adjusted OR =2.74; 95% CI = 1.18, 6.37, p=0.034).” Suggest: “However, having FD and regularly earning ≥THB 10,000 ($271.11 US) per month 30 was significantly associated with increased daily cigarettes use, after adjusting for age and 31 sex (adjusted OR =2.74; 95% CI = 1.18, 6.37, p=0.034).”

Also, clarify if analyses were only conducted among current smokers? The methods suggest this (“We 22 conducted a cross-sectional, nationwide and community-based study enrolling voluntary 23 participants aged ≥ 18 years who were current smokers….) but then the results suggest that this was a survey of current smokers and non-smokers (“Among 4597 respondents, 849 were current smokers…”). Also define what ‘current smoker’ means – how was this operationalized? And what was the outcome then – daily smoking status among the sample of current smokers?

The authors cannot make this statement – unless they also looked at smoking frequency and not just daily cigarette smoking status: “FD alone didn’t predict 32 increasing smoking frequency.”

Introduction:

Clarify this statement: “Another study found that the mean 53 number of daily cigarettes smoked was higher among substance-dependent unemployed 54 workers during the pandemic [13].”

The introduction could generally benefit from more thoughtful structure of content and flow from paragraph to paragraph. The structure and flow is not clear or intuitive in its current form, as it includes one paragraph that summarizes a range of points and then jumps to the research aim in paragraph 2. I could imagine a structure with the following paragraphs: 1) tobacco use as a public health problem associated with stress; 2) covid-19 and related psychological and financial distress; 3) prior research and mixed findings with regard to covid-19, financial distress, and smoking; 4) why studying this topic in Thailand is important; and 5) gaps and research aims.

Methods:

Could the authors reference the source for this: “We retrieved the secondary data from a nationwide study of the third-wave COVID-19 pandemic-related psycho-socio-economic characteristics to enroll the study participants from June to September 2021.” It isn’t clear how this fits? Is this the survey data analyzed? The authors state that “We enrolled the study participants…” Clarify please?

The measures section could be clarified and presented more consisely. For example, suggestion: “Participants were asked to report their employment status during the third wave of the COVID-19 pandemic: Came to work as usual; Quit working, Reduced number of work days, Temporary suspension of employment, Laid off or the termination of business, Dismissed from work, Contract Expiration, or Business Closure. The first two responses were defined as no FD; any of the subsequent responses were defined as FD.”

Please provide the exact language for this item, as it is not clear to me what the question was and the response options: “The participants who were current smokers would report the self-estimated number 109 of daily cigarettes smoked before and during the COVID-19 pandemic recession”.

Section 2.6 should come before Section 2.3-2.5 – and Sections 2.3-2.5 should be considered as part of (contained within) Section 2.6 or as a separate section including them all labeled “Measures” (in my opinion).

Results:

The table includes former smoker status, but this is not included as a measure in the methods.

Discussion:

In general, the discussion could be made more succinct and, like the introduction, could benefit from some reorganization for content, structure, and flow.

“Among the people who experienced FD and were high earners significantly increased number of cigarettes smoked during the third wave of COVID-19 pandemic recession in Thailand.” – Rephrase for grammar/sentence structure, e.g., “People in Thailand who experienced FD and were high earners reported significantly increased number of cigarettes smoked during the third wave of COVID-19 pandemic recession.”

Comments on the Quality of English Language

See above.

Author Response

(The authors gave the same response as above.)

Round 2

Reviewer 3 Report

Comments and Suggestions for Authors

The authors were generally responsive to the prior set of reviews and have improved clarity.